# Targeting White Adipose Tissue with Exercise or Bariatric Surgery as Therapeutic Strategies in Obesity

**DOI:** 10.3390/biology8010016

**Published:** 2019-03-15

**Authors:** Flávia Giolo De Carvalho, Lauren M. Sparks

**Affiliations:** 1School of Physical Education and Sport of Ribeirao Preto, University of Sao Paulo, Avenida Bandeirantes 3900, Ribeirao Preto, SP 14040-907, Brazil; flaviagiolo@gmail.com; 2Translational Research Institute for Metabolism and Diabetes, Advent Health, 301 East Princeton Street, Orlando, FL 32804, USA

**Keywords:** adipose tissue, obesity, exercise, endurance, resistance, human

## Abstract

Adipose tissue is critical to whole-body energy metabolism and has become recognized as a bona fide endocrine organ rather than an inert lipid reservoir. As such, adipose tissue is dynamic in its ability to secrete cytokines, free fatty acids, lipokines, hormones and other factors in response to changes in environmental stimuli such as feeding, fasting and exercise. While excess adipose tissue, as in the case of obesity, is associated with metabolic complications, mass itself is not the only culprit in obesity-driven metabolic abnormalities, highlighting the importance of healthy and metabolically adaptable adipose tissue. In this review, we discuss the fundamental cellular processes of adipose tissue that become perturbed in obesity and the impact of exercise on these processes. While both endurance and resistance exercise can promote positive physiological adaptations in adipose tissue, endurance exercise has a more documented role in remodeling adipocytes, increasing adipokine secretion and fatty acid mobilization and oxidation during post-exercise compared with resistance exercise. Exercise is considered a viable therapeutic strategy for the treatment of obesity to optimize body composition, in particular as an adjuvant therapy to bariatric surgery; however, there is a gap in knowledge of the molecular underpinnings of these exercise-induced adaptations, which could provide more insight and opportunity for precision-based treatment strategies.

## 1. Introduction

The prevalence of overweight and obesity has increased over three decades, reaching over 2 billion individuals worldwide, with a higher proportion among women [1]. A sobering statistic is that, in 2030, 51% of the population will be obese, and 10% of Americans will have a BMI over 40 kg/m^2^ [2]. The obesity epidemic has a major impact on public health, and the health problems associated with obesity are modulated by both environmental and genetic factors [3]. While excess adipose tissue (obesity) is associated with metabolic complications, a normal-weight healthy woman can have as much adipose tissue as an obese man with type 2 diabetes [4]. The adipose tissue mass is, therefore, not the only culprit in obesity-driven metabolic abnormalities, highlighting the importance of healthy and metabolically adaptable adipose tissue and not simply focusing treatment strategies on reducing mass alone. Increases in body weight (obesity) are directly related to adipose tissue expansion but are also characterized by perturbations in key adipocyte-specific genes that influence metabolism. Adipose tissue is a complex organ responsible for the generation and storage of neutral lipids (through lipogenesis, esterification), release of glycerol and free fatty acids (through lipolysis), as well as a number of other hormones, peptides, etc.—all of which require ATP; thus, understanding the molecular etiology of these obesity-driven metabolic perturbations will largely inform therapeutic strategies, such as exercise, that can be used to reduce its impact but also presents an important medical challenge [5].

The primary effect of exercise on adipose tissue is the catecholamine-induced release of free fatty acids as a fuel source for contracting skeletal muscles and other tissues. In fat cells, exercise training induces enzymatic adaptations promoted by the sympathetic nervous system in response to the depletion of triglycerides [6]. The effects of exercise specifically on white adipose tissue were first investigated in 1991 in rats [7]. A 10-week swimming protocol increased the activity of enzymes of the respiratory chain (cytochrome-c oxidase and malate dehydrogenase) in both white adipose tissue and skeletal muscle, showing a positive effect of endurance exercise on white adipose tissue metabolism in rats. While endurance and resistance exercise both promote positive physiological adaptations in adipose tissue, endurance exercise has a more documented role in remodeling adipocytes, increasing fatty acid mobilization and oxidation during and post-exercise, modulating adipokine secretion and regulating mitochondrial metabolism [8,9] compared with resistance exercise. Very little is known about the molecular transducers of the physiological responses to both endurance and resistance exercise in adipose tissue in both healthy and diseased populations. Exercise is considered a viable therapeutic strategy for the treatment of obesity, in particular as an adjuvant therapy to bariatric surgery and some pharmacotherapies, to optimize body composition and glycemic control [10]. As such, gaining a better understanding of the molecular pathways that are activated in response to exercise and how these might differ by training modality, e.g., endurance vs. resistance, will advance the pursuit of viable targets for deeper exploration as treatments for metabolic diseases such as obesity and type 2 diabetes.

## 2. Mitochondrial Content and Capacity

The capacity of adipose tissue to regulate energy metabolism is directly related to mitochondrial content. Mitochondria are responsible for generating adenosine triphosphate (ATP), but these organelles also produce reactive oxygen species (ROS) and are integral to the differentiation and maturation of adipocytes, which is modulated by the content of mtDNA and mtDNA-encoded components of the oxidative phosphorylation system in the adipose tissue [11]. It is important to note that the availability of ATP is also essential for the fundamental cellular processes that take place within adipocytes, such as lipolysis, esterification and lipogenesis [12]. In particular, adipocyte mitochondria regulate lipogenesis by also providing key intermediates such as glycerol 3-phosphate, acetyl-CoA and pyruvate for the synthesis of triglycerides [13,14], in addition to the regulation of lipolysis and fatty acid re-esterification [15].

Increasing body mass index (BMI) is associated with decreasing mitochondrial content [14], suggesting that as adipose tissue expands—as in the case of obesity—the ability of adipose tissue to generate a sufficient amount of energy to maintain its core metabolic processes is jeopardized. Meager evidence has demonstrated that exercise, particularly endurance, can potentially improve adipose tissue mitochondrial capacity through increasing mitochondrial content and the expressions of proteins related to mitochondrial biogenesis and metabolism (*PGC-1α*, uncoupling proteins, irisin, etc.) [16,17]. An even greater void exists in the literature regarding the effects of resistance exercise on the modulation of adipocyte mitochondrial content, capacity and the expressions of key metabolic genes such as uncoupling proteins (UCPs).

While rodent white adipose tissue can undergo robust browning and develop functional UCPs in response to exercise, these results have not been recapitulated in human white adipose tissue [8,18,19,20]. One potential explanation for these discrepancies is the significant difference in the anatomical distribution of adipose tissue between rodents and humans. As such, the adipocytes within these depots display metabolic heterogeneity. Human subcutaneous adipose tissue is continuous with dermal adipose tissue, while rodent subcutaneous adipose tissue is separated from dermal adipose tissue by a smooth muscle layer [18].

Although our focus in this review is white adipose tissue, another explanation for the difference between animal and human adipose tissue energy regulation is associated with brown adipose tissue, specifically related to the functionality, gene, and protein expressions [21]. In rodents, the brown adipose tissue depots are present in well-defined anatomical locations and are homogeneously composed of brown adipocytes, whereas human brown adipose tissue is dispersed throughout the body in the neck, vertebral and peri-renal areas and composed of a mixture of white, brown, and brite adipocytes [21,22]. Additionally, when analyzing UCP-1 expression, the similarity between mouse and human UCP1 is less than 80%, which suggests a possible functional difference between species [23].

Furthermore, a previous study developed by our research group [19] investigated the effects of a 3-week training protocol on markers of mitochondrial respiration and content in the adipose tissue in lean and overweight sedentary individuals. It was found that the training protocol effectively increased the aerobic capacity (maximal phosphorylation capacity (ATPmax) and maximal oxygen uptake (VO2peak)), but had no impact on mtDNA content and the expressions of thermogenic and oxidative genes in white adipose tissue [19].

The results of Pino et al. study [19] showed that the benefits of exercise on adipose tissue go further then changing mitochondrial content, since that no changes were observed in the mtDNA content, and the mRNA levels of energy regulator genes (*UCP1*, *PRD1-BF1-RIZ1 homologous domain containing 16* (*PRDM16*), *peroxisome proliferator-activated receptor gamma coactivator 1-alpha* (*PGC1a*), and *carnitine palmitoyltransferase 1B* (*CPT1b*)) in the white adipose tissue were unaffected by the training protocol. However, functional metabolic improvement occurred since that important clinical changes in body composition (fat mass decrease) and aerobic capacity markers (increased levels of ATPmax and VO2peak) were observed. The ATPmax of the lean/overweight sedentary participants increased to a level similar to the active participants, showing an improvement in aerobic capacity.

In addition, differences also exist between rodents and humans in the pathways that drive lipolysis [18]. In the catecholamine-driven pathway, lipolysis is modulated by the activation of three different β-adrenergic receptor subtypes (β1, β2, and β3), which leads to the activation of a cascade of events that includes the increased phosphorylation of cyclic adenosine monophosphate (cAMP) by adenylyl cyclase, activation of protein kinase A and activation of hormone-sensitive lipase, promoting its translocation to the lipid droplets to catalyze triglycerides hydrolysis [24]. One major difference between rodents and humans deals with β-adrenergic receptor (AR) expression. Both β1-AR and β2-AR are expressed in rodents and humans, whereas β3-AR expression is exclusive of rodent white adipose tissue and induces a lipolytic response in rodents. In humans, there is also the expression of α-adrenergic receptors in subcutaneous adipose tissue, which has a greater affinity for catecholamines and acts as a lipolytic inhibitor by decreasing cAMP levels. Thus, a balance between α2- and β-adrenergic receptors is required for lipolysis regulation only in humans [18]. There are also circulating factors that regulate lipolysis in human white adipose tissue such as growth hormone, cortisol, testosterone, ghrelin [25], and natriuretic peptide [26]. However, the effects of natriuretic peptide on lipolysis regulation is specific to human adipocytes and undocumented in rodent adipocytes. In summary, even though there are important similarities in the biology of rodent and human adipose tissue lipolysis, there are certain mediators of lipolysis in rodent adipocytes that simply do not have the same effects in human adipocytes [27]. These discrepancies should be considered when utilizing rodent models as a comparator for humans, especially when analyzing the response of adipose tissue specifically related to the lipolytic process.

White adipose tissue is malleable and responds to various endogenous (i.e., hormones, sexual dimorphism and aging) and exogenous stimulus such as food intake, exercise, stress and cold temperature [18]. Changes in human white adipose tissue in response to these stimuli, especially exercise, appear to be more related to the fundamental cellular processes within white adipose tissue such as lipolysis and oxidative capacity rather than a change in thermogenic capacity as has been observed in rodents. There are some specific characteristics and fundamental processes that are unique to the white adipose tissue depot, which exercise can perhaps enhance and hence improve whole-body metabolic health without having to turn white adipose tissue beige or brown by making it thermogenic. In other words, human white adipose tissue has the capacity to become more metabolically efficient and remain “white” rather than acquiring the characteristics of rodent brown adipose tissue and becoming more “beige” or brown. Future studies focused on lifestyle interventions and/or therapeutic agents aimed at increasing thermogenesis in human white adipose tissue (i.e., “browing/beiging”) might consider whether this is physiologically relevant or whether simply improving the metabolic flexibility of this organ, which is engineered for the storage and release of lipids as well as other signaling molecules, might be more advantageous for whole-body metabolic health.

## 3. Adipocyte Size and Number

The growth and differentiation of preadipocytes is controlled by communication between individual cells and/or between cells and the extracellular environment (hormones and growth factors), that affect adipocyte differentiation in a positive or negative manner. The differentiation is as important approach to reduce metabolic complications associated with obesity as evidenced by the adipose expandability hypothesis by Danforth [28]. The ability of the adipose tissue to expand—which requires pre-adipocyte differentiation into mature adipocytes—in the face of excess nutrients provides a “safe harbor” for otherwise deleterious fatty acids that would be ectopically stored in other organs that are not equipped to handle such metabolic burden. In addition, there are cellular components, such as preadipocyte factor-1 and extracellular matrix proteins, that can regulate the differentiation process and ultimately result in transcriptional activation, via *peroxisome proliferator-activated receptor γ* (*PPARγ*) and *CCAAT-enhancer-binding proteins* (*C/EBP*), and determinate the physiological and pathophysiological mechanisms underlying adipose tissue development and regulate lipid deposition [29].

The accumulation of adipose tissue is highly associated with overnutrition and low daily physical activity and exercise. In response to changes in energy status, adipose tissue is dynamically remodeled through modifications in the number and/or size of adipocytes and is closely associated with adipose tissue dysfunction [30]. Changes in the number (hyperplasia) and size (hypertrophy) of the adipocytes affect the microenvironment and under pathophysiological conditions, such as obesity, aberrant adipose tissue remodeling may induce a dysregulation in adipokine secretion, promote local hypoxia, and increase fatty acid fluxes, leading to metabolic stress and disorders [31,32].

Hyperplasia and hypertrophy are regulated by nutritional behavior and genetic factors; however, it remains unclear how adipose tissue remodeling is controlled at the molecular level [30,33]. In hyperplasia, adipocyte precursor cells differentiate into mature adipocytes by transcription factors that are regulators of adipogenesis, including *PPARγ* and *C/EBP* families and sterol regulatory element-binding transcription factor 1c (SREBP1c), which stimulates the expression of lipogenic genes, such as acetyl-CoA carboxylase, fatty acid synthase, and saturated fatty acid dehydrogenase [30].

Hyperplasia is related to higher levels of adiponectin and fewer inflammatory adipokines. Hypertrophic adipocytes, on the other hand, release less adiponectin and more inflammatory adipokines. The “flooding” of the adipocytes in the adipose tissue causes a reduction of the blood flow with consequent hypoxia and infiltration of macrophages, and increased lipolysis. Higher levels of free fatty acids in the plasma can lead to a lipotoxicity state [29,34]. In the face of positive energy balance and increased visceral fat mass, there is a spillover of excessive amounts of lipids which would promote ectopic lipid deposition and the release of inflammatory cytokines, resulting in lipotoxicity in adjacent tissues, such as liver and skeletal muscle, and ultimately leading to insulin resistance and impairments in glucose homeostasis [34,35]. Furthermore, the cytokines produced by macrophages such as tumor necrosis factor-alpha (TNFα), plasminogen activator inhibitor-1 (PAI-1), interleukin-6 (IL-6), retinol-binding protein 4, monocyte chemoattractant protein-1 (MCP-1) and acute phase proteins, can inhibit the adipogenesis process [36].

The balance of the adipocyte size expansion of mature adipocytes and adipogenesis directly affects metabolic health; in this sense, the smaller the adipocyte’s size, the lower the susceptibility of developing obesity, diabetes and other metabolic diseases, which supports the importance of increasing the number of pre-adipocytes and enhancing differentiation to provide an inert lipid store. Therefore, targeting adipocyte precursor cells, which ultimately impacts adipocyte differentiation, is an attractive treatment strategy for obesity and metabolic disease.

## 4. Secretory Function, Tissue Crosstalk and Inflammation

Adipose tissue is comprised of adipocytes that secrete numerous peptides and non-peptides, which have endocrine, paracrine and systemic actions and promote tissue crosstalk (Table 1). Adipose tissue also receives afferent signals from whole-body hormone systems, as well as the central nervous system, through receptors for insulin, glucagon, glucagon-like peptide-1, androgen and estrogen, cytokine and catecholamines, etc. [37,38].

The dysregulation of adipocyte-secreted factors is associated with perturbed mitochondrial metabolism in the face of excess adipose tissue mass (obesity) and can lead to pro-inflammatory states, insulin resistance, hyperglycemia, dyslipidemia, hypertension and the development of metabolic diseases [39]. Hypertrophic adipocytes display higher rates of basal lipolysis, increasing the release of free fatty acids into circulation. These fatty acids are taken up by other tissues such as liver and muscle and, when in excess, can promote ectopic lipid accumulation and lipotoxicity [40]. Furthermore, excess saturated fatty acids, especially palmitic or stearic acids, activate the signaling inflammatory cascade by the physical interaction of lipopolysaccharides and myeloid differentiation protein-2 (MD2), a co-receptor of Toll-like receptor 4, which promotes a dimerization interface and the formation of an active complex called TLR42–MD-2-2 (*toll-like receptor 4/myeloid differentiation factor 2 complex*) that reprograms macrophage metabolism [41], promoting chronic inflammation and insulin resistance, consequently disturbing energy metabolism [30].

Exercise, however, can modulate the immune–metabolic interface through the release of humoral factors that interact with adipose tissue. For example, during exercise myokines such as irisin, myostatin, myonectin, interleukins (IL) 6, 10 and 15 are produced and released by muscle fibers and present both local and pleiotropic effects. Myonectin and irisin [42] are related to glucose and fatty acid uptake and oxidation regulation in liver and adipose tissue. The crosstalk between skeletal muscle and adipose tissue can decrease adiposity, as well as increase thermogenesis—at least in some animal models. Even though those myokines play an important role in the regulation of energy metabolism, the mechanisms of action of the myokines alone or in concert remain unclear [43,44].

Low-grade systemic or “sterile” inflammation caused by obesity is associated with excessive adiposity and accompanied by local inflammation, fibrosis, vascular rarefaction and hypoxia in adipose tissue. Together, these events lead to disordered lipid metabolism, the dysregulation of adipocytokines, systemic insulin resistance and even cellular senescence [45]. Some inflammatory genes have recently been shown to be regulated through epigenetic mechanisms, thus highlighting the importance of environmental factors in promoting changes in the inflammatory profile of the adipose tissue [32,46]. Another critical point is that the inflammatory cytokines released by adipose tissue, such as interleukins (IL-6 and IL-1b), create a deleterious environment for skeletal muscle. Higher levels of IL-6 downregulate insulin-like growth factor/protein kinase B (IGF/Akt) signaling and decrease muscle protein synthesis by impairing insulin signaling in the myotubes and compromising myoblast differentiation/proliferation capacity [47], promoting muscle atrophy. As such, higher incident rates of obesity are associated with sarcopenia [42] and highlight the importance of exercise not only for body fat mass control, but also for the regulation of glucose and lipid metabolism and the prevention of muscle wasting and metabolic diseases, and even aging.

However, it is important to point out that IL-6, which is an exercise-inducible cytokine in both humans [48] and rodents [49], can also promote positive effects on glucose metabolism. A recent study showed that the injection of IL-6 improved hepatic glucose and insulin homeostasis in vivo due to an indirect effect on the hepatocyte. The authors investigated the effects of a low fat diet (10% kcal) vs. a high fat diet (60% kcal) for seven weeks associated with the injection of IL-6 (400 ng, 75 min) in mice and it was observed that IL-6 decreased blood glucose levels and increased the phosphorylation of AKT through the mRNA expression of gluconeogenic genes, without changing circulating insulin levels [50]. In addition, there is some evidence that the capacity of IL-6 to regulate energy metabolism is related to AMP-activated protein kinase (AMPK) in animal models [51]. AMPK is an energy sensor protein and its phosphorylation activates fatty acid oxidation and increases glucose uptake, and it can be modulated by IL-6 in both muscle and adipose tissue in mice [52].

Besides metabolism regulation, IL-6 can have anti-inflammatory effects related to its ability to stimulate the production of interleukins (IL-1 receptor antagonist and IL-10) that modulate the production of pro-inflammatory factors (IL-1α, IL-1β and TNFα) and chemokines (IL-8 and macrophage inflammatory protein α) from lipopolysaccharide-activated in human macrophages [51]. Even though there is evidence for a beneficial role of IL-6 in hepatic cells, the benefits of IL-6 in adipose tissue need to be further investigated. It is important to consider that a detrimental role of IL-6 exists [37,42] and that obesity particularly increases IL-6 levels both in humans and mice due to macrophage secretion in the adipose tissue [50], promoting a low-grade systemic inflammation that can impair insulin signaling and glucose metabolism [45]. However, elevated IL-6 levels may be more detrimental in combination with low physical activity rather than simply obesity alone. The metabolic context of the levels of these cytokines should therefore be considered when determining their role in metabolic health.

## 5. White Adipose Tissue and Exercise: Endurance vs. Resistance Training

It is widely accepted that exercise stimulates mitochondrial biogenesis, thereby increasing mitochondrial density and oxidative phosphorylation capacity in skeletal muscle [44]. Exercise-induced mitochondrial biogenesis also occurs in other tissues including brain, liver and adipose [53], demonstrating that exercise training increases metabolic requirements in these tissues [16]. Much like skeletal muscle, adipose tissue releases proteins (and other factors) into the circulation in response to exercise that modulate metabolism in other peripheral tissues, suggesting that adipose tissue may also have the ability to direct inter-organ crosstalk and not only serve as a target tissue for other exercise-responsive organs [54].

How does endurance vs. resistance training affect adipose tissue? Muscle contraction induces the release of myokines that are exercise modality-specific and have paracrine effects on other tissues, such as adipose [55]. For example, endurance exercise promotes the release of factors such as *fibronectin type III domain-containing protein 5* (FNDC5)/Irisin, IL-15, IL-6 and myostatin, while resistance exercise is linked with the release of myonectin, b-aminoisobutyric acid (ANGPTL4), b-aminoisobutyric acid (BAIBA) and fibroblast growth factor 21 (FGF21). Even though the secretory profiles differ by modality, the exercise-induced myokines are involved in the metabolic improvements in adipose tissue such as reduced inflammation, increased glucose uptake and, in some rodent models, an increase in thermogenesis (i.e., “browning”) of the white adipose tissue [56]. Furthermore, endurance exercise enhances the catecholamine response in adipose tissue, thereby improving fatty acid mobilization via lipolysis in order to support fatty acid delivery to skeletal muscle [8]. These adaptations are associated with marked changes in the gene expressions of *PPARγ*, particularly related to oxidative phosphorylation [57], that are activated by the adenosine monophosphate-activated protein kinase (AMPK) and sirtuin 1 (SIRT1). Both AMPK and SIRT1 work as sensors of energy status and are regulated by energy availability (low levels of adenosine triphosphate (ATP)), such as starvation or during exercise [8,9]. Upon activation, *PPARγ* increases the expression of *FNDC5* and, consequently, the release of an adipokine called irisin, which can promote thermogenesis, or “beiging/browning”, of white adipose tissue—at least in rodent models [30,58]. In addition to the direct effects of the exercise-induced/catecholamine-stimulated release of free fatty acids, exercise has been associated with improvements in the insulin suppression of free fatty acids in the post-prandial state (i.e., after feeding). The ability of adipose tissue to respond to exercise-/fasting-induced catecholamine signals, as well as insulin signaling in the fed state, is critical to the health of adipose tissue and the fundamental cellular responsibilities of this organ [59].

Other studies have shown the positive effects of exercise on adipose tissue mitochondrial metabolism in rodents [60,61,62,63]. A recent report demonstrated that endurance exercise increased the content of mitochondrial proteins (complex proteins of cytochrome c oxidase subunit IV) and cytochrome c oxidoreductase core I subunit in addition to increased transcript levels of *PPARγ*, *mitochondrial transcription factor A* (*TFAM*) and citrate synthase activity, enhancing mitochondrial function [61]. Resistance exercise has been shown to stimulate the release of a hormone called meteorin-like from skeletal muscle, which in turn increased energy expenditure, improved glucose tolerance and elevated thermogenesis, specifically in white adipose tissue in obese mice [62]. The metabolic consequences of this hormone were not directly related to changes in thermogenic gene expression but rather with the stimulation of immune cell subtypes that activate thermogenic actions in adipose tissue, possibly by stimulating the expression of the active macrophages (M2) and other regulatory cytokines such as IL-10, IL-4 and IL-13. A recent study [60] evaluated the effects of 10 weeks of endurance exercise with or without whey protein supplementation on visceral adipose tissue in rats fed a high fat diet. They observed that the high fat diet promoted weight gain and increased the body adiposity index, glucose levels and inflammatory markers compared to the normal diet. However, only the combination of endurance exercise and whey protein reduced the fat pads and local levels of inflammatory markers (TNF-α, hypoxia-inducible transcription factors (HIF-1α), vascular endothelial growth factor A (VEGF-A)) in visceral adipose tissue [60]. Taken together, these studies demonstrate that exercise robustly impacts mitochondrial gene expression and activity in white adipose tissue, and these effects occur in response to different durations and modalities of exercise [19].

In contrast to the findings in animals, studies in humans have shown that endurance exercise has no impact on mitochondrial metabolism in white adipose tissue [19,64,65,66]. A previous study investigated the effects of a 3-week training protocol on markers of mitochondrial respiration and content in the adipose tissue. It was found that the training protocol effectively increased the aerobic capacity (ATPmax and VO2peak) but had no impact on mtDNA content and the expressions of thermogenic and oxidative genes in white adipose tissue, as mentioned in the “Mitochondrial Content and Capacity” session [19].

A 12-week progressive aerobic and resistance exercise program evaluated markers of lipolysis, inflammation, adipokines and mitochondrial function in abdominal subcutaneous adipose tissue biopsies of obese men [64]. The authors observed that the exercise program enhanced oxygen consumption and strength (1 repetition maximum (1RM)), as well as peripheral insulin sensitivity, following the exercise intervention. However, there were no changes in the protein expressions of *PGC-1* α, markers of lipolysis (genes *adipose triglyceride lipase* (*ATGL*), *hormone-sensitive lipase* (*HSL*), *comparative gene identification-58* (*CGI-58*), *perilipin 1*) and other markers related to adipose tissue inflammation (TNFα, IL-6, MCP-1, *cluster of differentiation 68* (CD68)), browning (*cell death-inducing DNA fragmentation factor alpha-like effector A* (CIDEA), PRDM16), mitochondrial biogenesis (*PGC-1α*) and adipokine expression (*autosomal dominant polycystic kidney disease* (*ADPK*), *leptin* (*LEP*)). In accordance with these reports discussed above [19,64], another study [65] investigated the effects of high-intensity interval exercise training on mitochondrial fat oxidation in skeletal muscle and adipose tissue in overweight subjects and observed that, despite promoting metabolic adaptations in skeletal muscle, ten days to 6 weeks of high-intensity interval exercise training did not increase white adipose tissue oxidative capacity in healthy individuals. Another study conducted in sedentary men and women with either normal glucose tolerance, impaired glucose tolerance or type 2 diabetes examined the effects of a 4-week exercise program (three 1-h sessions of swimming or biking per week). This exercise regimen increased the gene expressions of *PGC-1α* and *PPARγ* in adipose tissue in all participants [67]. Regarding mitochondrial activity, a recent study investigated the effects of a 6-month supervised endurance exercise intervention on mitochondrial activity in healthy men with low levels of physical activity. Following the 6-month endurance exercise intervention, the expressions of genes involved in oxidative phosphorylation and proteasome activity were increased in adipose tissue, indicating an elevated protein turnover. Since adipogenesis is a continuous process, a higher protein turnover may reflect increased metabolic activity in adipose tissue in response to exercise [57].

It has been hypothesized that exercise-induced browning/beiging of white adipose tissue occurs due to exercise, decreasing adipocyte lipid content and size and consequently decreasing insulation of the body, demanding a higher heat production, thus inducing “browning/beiging” of the subcutaneous white adipose depot [68,69,70]. Current findings in the literature are somewhat contradictory and inconclusive in humans. A recent study implemented 12 weeks of cycling (3 times per week, intensity 70–80% HRmax) in sedentary men and women and found increases in circulating adiponectin, apelin and irisin levels despite no changes in body composition. Interestingly, the gene expressions of UCP1, *T-box transcription factor 1* (*TBX1*) and *carnitine palmitoyltransferase 1B* (*CPT1B*) were significantly increased in subcutaneous white adipose tissue [71]. Short-term cold exposure in endurance-trained athletes and lean sedentary males increased gene expression of *FNDC5* in skeletal muscle in endurance athletes but plasma irisin and IL-6 levels were similar in both groups. Furthermore, no differences in *UCP1* and *PGC-1α* gene expressions, and brown and beige adipocyte markers (*Cidea, TMEM26, CD137*), were detected in subcutaneous white adipose tissue between the trained and sedentary subjects [66].

Despite no effect on mitochondrial protein expressions, a 4-month aerobic exercise program (five days/week) in healthy young men demonstrated enhanced lipid mobilization, whereby the participants had reduced plasma non-esterified fatty acid levels at rest and during exercise with elevated lipid oxidation and lipolysis, at rest, during exercise and following an exercise bout [67]. Exercise-induced lipid mobilization occurs through the upregulation of genes related to phospholipid metabolism (*AGPAT3*, *Glycerol-3-Phosphate Dehydrogenase 1* (*GPD1*), *PLA2G12A, Glycerol-3-Phosphate Acyltransferase* (*GPAM*), *IPLA2G*) and genes that regulate fatty acid biosynthesis and elongation (*EVOL3*, *EVOLU4*, *ACACA*, *GPAM*, *AGPAT3*, *PPARγ*), which, combined, promote a decrease in triacylglycerol, phosphatidylserines, lysophosphatidylglycerols and lysophosphatidylinositols contents in adipose tissue [68]. Furthermore, the effects of exercise on the structural lipids of subcutaneous adipose tissue were investigated using shotgun lipidomics and found that as little as three weeks of exercise were sufficient to decrease phosphatidylserines (PSs), lysophosphatidylglycerols (LPGs), lysophosphatidylinositols (LPIs), and triacylglycerols (TAGs) and upregulate genes involved in phospholipid metabolism (*AGPAT3*, *GPD1*, *PLA2G12A*, *GPAM*, *IPLA2G*) [72]. In this context, adipose tissue is an immediate fuel source for the working muscle cells during exercise, but it can also contribute to longer-term reductions in body fat mass. The important point is that, even though there is clear evidence for specific changes promoted by exercise on mitochondrial adipose tissue metabolism, the changes in lipid content promoted by exercise reflect a selective remodeling in adipose tissue that has significant implications for other basic cellular processes of adipose tissue such as adipogenesis and glucose and lipid handling. Epigenetic adaptations can occur in response to exercise, such as the gene-specific DNA hypomethylation of *PGC-1α*, *PPARγ*, *TFAM*, pyruvate dehydrogenase lipoamide kinase isozyme 4 (PDK4) and citrate synthase [69]. A recent report demonstrated that transcriptomic and DNA methylation changes were induced with both acute and chronic exercise training, specifically related to immune inflammatory pathways [70].

There are numerous reports on the effects of endurance exercise on adipose tissue; however, meager evidence exists for the effects of resistance exercise on adipose tissue. Resistance exercise works by different metabolic pathways—specifically, it has a more profound effect on muscle mass through the stimulation of protein synthesis via the *mammalian target of rapamycin* (mTOR) pathway [73], and the main substrate utilized during exercise is glucose via ATP in skeletal muscle. Thus, it is tempting to speculate that the effects of resistance exercise on adipose tissue might be minor; however, this would be premature. Most of the studies published have narrowly examined the effects of resistance exercise on body composition or serum lipid modulation [59,72,73,74]. Resistance exercise can modulate lipid metabolism by increasing muscle oxidative capacity and intramyocellular triglyceride utilization. A 6-week resistance training protocol was conducted in sedentary men and found adaptations that were similar to endurance exercise regarding lipid metabolism. The study observed increased oxidative capacity (measured as cytochrome c oxidase protein content), intramyocellular triglyceride and lipid droplet-associated perilipin 2 and perilipin 5 content in Type I and II muscle fibers [74]. Therefore, besides increasing muscle mass, resistance exercise would be a viable strategy for improving local skeletal muscle lipid metabolism and perhaps reduce the risk of metabolic disease in obese and sedentary individuals. These same aspects of local lipid metabolism in response to resistance exercise have not been thoroughly investigated in adipose tissue.

Studies have shown positive results in relation to resistance exercise and body composition as increasing isometric lean mass and decreasing fat mass with a resistance exercise protocol of five months [75]. This same 5-month exercise study also evaluated the effects of combined endurance and resistance exercise and found improvements in markers of lipid metabolism (apolipoproteins A-1 and B), in addition to the observed changes in body composition [75]. These results highlight the important effects of resistance exercise on muscle mass and its indirect effects on pathways related to adipose tissue. The dynamics of a 3-month resistance exercise protocol on the control of lipolysis was investigated in obese men [59]. The authors observed that the resistance exercise increased the responsiveness to the beta-adrenergic receptor stimulation of lipolysis and anti-lipolytic action of catecholamines via alpha-adrenergic 2A receptor (ADRA2As). In addition, whole-body and adipose tissue insulin responsiveness was improved by the resistance exercise protocol. Eight weeks of intense resistance training in health young men [72] found increased lean mass and improved fatty acid utilization—potentially through the increased expressions of the genes encoding *ADIPOR1* and *Cytochrome C Oxidase* (COX4)—in skeletal muscle; however, no changes were observed in the oxidative metabolism genes (*PGC-1 α*, *SIRT1*, *TFAM*, *CPT1b*, and *FNDC5*) in skeletal muscle or adipose tissue (*AdipoR1*, *COX4* expression). Therefore, resistance training seems to improve the regulation of lipid mobilization in abdominal subcutaneous adipose tissue and can possibly modify the state of metabolic inflexibility in adipose tissue in obese subjects [59], although this has not been demonstrated in large cohorts and/or through multiple studies.

While both endurance and resistance exercise promote physiological and epigenetic adaptations in adipose tissue, endurance training appears to be more effective in decreasing body fat mass and adipocyte size, increasing fatty acid mobilization and oxidation during and post-exercise, and modulating adipokine secretion [8]. The effects of resistance exercise specifically related to adipose tissue metabolism are still unclear but resistance exercise appears to largely impact fatty acid mobilization during exercise and for numerous days following exercise.

Adipose tissue serves an important purpose in the sequestering of neutral lipid and its fundamental cellular responsibilities are to maintain lipid flux for the whole body. In this way, exercise can help to maintain and/or improve these processes by increasing energy expenditure and lipid substrate utilization, and modulating myokine release. As we have highlighted in this review, some evidence exists to support the notion that positive adaptations in the metabolic efficiency of white adipose tissue can occur in response to exercise of either training modality (i.e., endurance or resistance). However, the mechanisms through which exercise affects each aspect of these fundamental cellular processes of white adipose tissue is unclear, especially with regard to resistance training. Since the primary function of white adipose tissue is to store energy [5], it will only become metabolically active in response to increasing energy demand in order to provide substrates for other working tissues, such as skeletal muscle [8]. It is tempting to speculate, therefore, that the beneficial effects of exercise are likely to be reaped when demand is consistent, as in performing repeated bouts of exercise for long periods of time, and this supports the therapeutic strategy of adopting some form of an exercise regimen as a lifestyle modification.

## 6. Exercise, Bariatric Surgery and Adipose Tissue

Among the treatment options for obesity, bariatric surgery is one of the most effective options, and studies have demonstrated the metabolic benefits such as increased insulin sensitivity, modulation of adipokine secretion and decreased local adipose and systemic inflammation [76]. Obese individuals typically have high levels of endoplasmic reticulum (ER) stress, and [77] high levels of ER stress are significantly reduced one year post-bariatric surgery. In addition, the weight loss induced with bariatric surgery decreases the infiltration of macrophages and secretion of pro-inflammatory molecules in adipose tissue [78].

Recent studies have shown that bariatric surgery also has a metabolic impact at the epigenetic level [79]. For example, intense weight loss and caloric restriction associated with bariatric surgery promote important changes in the DNA methylation of key genes within adipose tissue (*CETP*, *DNMT3B*, *FOXP2*, *HDAC4*, *KCNQ1*, *HOX* clusters). However, the functional relevance of these changes at the level of DNA methylation have yet to be determined. Future studies focused on the epigenetic regulation of adipocyte function following dramatic weight loss induced by bariatric surgery are on the horizon. Bariatric surgery not only remodels adipocyte function but also the interaction between adipose tissue and other organs. For example, Roux-en-Y gastric bypass bariatric surgery has dramatic effects on the serum and adipose tissue levels of adiponectin and inflammatory markers (MCP-1 and TNF-a) [80]. Even prior to the weight loss, a significant increase in adiponectin levels in adipose tissue and a decrease in serum MCP-1 levels have been observed as early as two weeks following surgery, demonstrating reduced inflammation without a reduction in fat mass.

In relation to adipose tissue mitochondrial metabolism, a recent study [81] evaluated adipose tissue mitochondrial respiration and lipolysis following massive weight loss promoted by diet and Roux-en-Y gastric bypass in obese patients with or without type 2 diabetes. It was noted that the mass-specific respiratory capacity of adipose tissue increased with weight loss in all patients, and the mitochondrial respiratory rates increased and were similar in both groups. Also, the ratio between the maximal capacity of the electron transport system and oxidative phosphorylation system capacity, lipolysis and insulin sensitivity increased 18 months after the surgery in both groups. When evaluating the respiratory capacity of adipose tissue following such a dramatic loss of adipose mass, the way in which these measurements are normalized becomes critical. For example, mitochondrial content should be considered in the event that the weight loss reduces mitochondrial content. We know from previous studies that mitochondrial content actually decreases with increasing body mass index and fat mass. Furthermore, previous studies have shown dramatic differences in respiration between human white and brown adipocytes due to differences in mitochondrial content [82,83]. Accounting for lipid droplet content is also another aspect to consider when quantifying respiration in adipose in the event of such dramatic weight reduction, since weight loss inevitably affects lipid droplet size and number. One way in which this has been done is by measuring the content of the lipid droplet surface protein perilipin 1 [84,85]. Furthermore, bariatric surgery can also promote skeletal muscle mitochondrial metabolism [14]. The effects of Roux-en-Y gastric bypass surgery and exercise were verified in 101 patients who underwent RYGB surgery and completed either a 6-month moderate exercise or a health education counseling intervention [10]. The authors observed that both groups presented similar reductions in body weight, fat mass, BMI and waist circumference after the interventions; however, endurance exercise provided additional improvements in insulin sensitivity, cardiorespiratory fitness, skeletal muscle mitochondrial respiration and enzyme activities, without changing mitochondrial content [10].

In general, bariatric surgery elicits an intense weight loss of both fat and fat-free (i.e., muscle) mass. Even though the benefits observed in relation to exercise post-surgery, there are currently no exercise or physical activity guidelines specifically for bariatric surgery patients. However, the American Society for Metabolic and Bariatric Surgery (ASMBS) and the American Heart Association (AHA) recommend that patients adhere to a healthful lifestyle and exercise for at least 30 min per day, but the modality, frequency and intensity have not been investigated or recommended [86]. Therefore, bariatric surgery is an effective treatment for obesity, especially considering the capacity to modulate weight loss, as well as the systemic improvements in insulin responsiveness for the potential treatment of insulin resistance and type 2 diabetes. Based on limited data, when combined with exercise, the benefits of bariatric surgery are amplified, specifically through the enhancement of mitochondrial metabolism and cardiorespiratory fitness. Exercise thus potentially offers an important strategy for those patients who experience suboptimal weight loss or experience weight regain in the years post-surgery.

Simply eliminating the adipose tissue (i.e., weight loss, bariatric surgery) is not the only solution to improving metabolic health. In fact, a reduction of adipose tissue excess can certainly be beneficial, but it is not obtained through exercise alone and usually requires some level of caloric restriction. Changes in lifestyle such as consistent exercise and balanced food intake are essential to promote a metabolically “fit” adipose tissue.

## 7. Conclusions

The importance of adipose tissue to the whole-body energy metabolism is undeniable; however the impact of endurance or resistance exercise on adipose tissue dynamics remains largely unknown and understudied, especially in the context of obesity-driven metabolic disease. Much of the early work in this area of exercise and adipose tissue has focused on understanding the organ’s ability to secrete cytokines, free fatty acids, lipokines and hormones in response to acute and chronic exercise, as well as lipid handling within adipocytes. In this exciting time of technological advancements toward understanding the molecular transducers of response, studies employing endurance and resistance training interventions with adipose and muscle biopsies offer the opportunity to better understand how adipose tissue may direct inter-organ crosstalk and its capacity to modulate metabolism in other peripheral tissues. Furthermore, the combination of bariatric surgery and exercise can amplify the metabolic benefits not only through reductions in adipose mass, but perhaps through remodeling its fundamental cellular process such as lipid turnover and mitochondrial metabolism. Efforts should be continued to elucidate the mechanisms and potential effects of exercise on white adipose tissue as a therapeutic strategy in obesity.

## Figures and Tables

**Table 1 biology-08-00016-t001:** Adipocyte-secreted peptides and non-peptides, and related endocrine/paracrine and systemic actions.

Endocrine/Paracrine and Systemic Actions	Adipocyte-Secreted Peptides and Non-Peptides
Endocrine function	Adiponectin, Leptin, angiotensin, estrogen and androgen hormones, insulin growth-like factor-1 (IGF-1)
Molecules related to lipid metabolism or transport	Lipoprotein lipase, cholesterol ester transfer protein, apolipoprotein E, non-esterified fatty acids (NEFAs)
Cytokines and immune-related proteins	TNF-alfa, IL-6, MCP-1, adipsin, acylation-stimulating protein (ASP), resistin, visfatin

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
