# Peer review of "Targeting White Adipose Tissue with Exercise or Bariatric Surgery as Therapeutic Strategies in Obesity"

_biology, 2019, doi:10.3390/biology8010016_

Reviewer 1 Report

The authors discuss the role of white adipose tissue (WAT) in the metabolic complications accompanying obesity - and in particular the therapeutic potential of exercise to restore WAT function the context of obesity.

The article is clearly written and easy to follow. With that said, i am not particularly enthusiastic about the content of the article. It has the feel of a somewhat superficial literature review that offers little in terms of an new slant or opinion on the current state of the field.

To me, it reads a little like a review article written for the sake of it. It lacks focus, fails to really underscore controversy in the literature, or offer clear opinion on where the field needs to go. 

My specific comments are listed below:

In the abstract you state that 'In this review....'. Im not really sure you actually discussed these fundamental processes in great depth. I would suggest revising this statement of your objective here - and also repeat it toward the end of the introduction section. The statement have at the end of the intro section currently reads like the lead-in to an aim and hypothesis for an original article, where a clear statement of the purpose of your review article would be more preferable.

Related to point one - given that you devote a good amount of space to the impact of bariatric surgery WAT, perhaps your title could be changed to reflect this? i.e., drop the 'with exercise' in the title.

Many of the sections describe some studies, highlight some inconsistencies in the literature, and then conclude that more work is needed. I urge the authors to offer their opinion of why they think there may be a discord in the literature is some areas of WAT metabolism, and to offer their opinion on where the field should go.

I would encourage you to provide a more robust discussion of the impact of exercise on WAT, and in particular, WAT browning. While there is a growing body of literature on this topic derived from rodent studies - studies in humans are fewer and far less compelling. While rodent WAT can undergo robust browning - developing functional UCP1 - to my knowledge there are no reports to date showing that human WAT has the capacity develop functional UCP1. Perhaps this is the issue - rodent and human WAT are fundamentally different in terms of their ability to undergo browning? I urge you to make your case as to why you think rodent and human studies do not align on this matter - what could be done to reconcile this issue in translatability?

So when mito respiration increases in WAT with weight loss or exercise, how is respiration being normalized? Do you think mito volume/density is increasing per adipocyte? Or is a significant shrinking of the adipocyte lipid droplets responsible for this observation. What is a good marker for mito content in WAT?

Your description of recent bariatric surgery data is interesting. Given the impressive - and perhaps most critically - the sustained weight loss achieved by sleeve gastrectomy and bypass procedures, I wonder if these findings reflect what happens in WAT following chronic and sustained weight loss? Whereas perhaps many exercise studies really only reflect and more acute response?

Perhaps it would also be interesting to comment on what real-world value of surgery vs exercise might be given that exercise interventions rarely provide long-term sustained weight-loss.

Author Response

We would like to thank the Editor and reviewers for their attention and review of our manuscript. All of the reviewers' comments were carefully considered and addressed, and we feel that the manuscript has significantly improved as a result. All changes are indicated with red underline.

The authors discuss the role of white adipose tissue (WAT) in the metabolic complications accompanying obesity - and in particular the therapeutic potential of exercise to restore WAT function the context of obesity.

The article is clearly written and easy to follow. With that said, I am not particularly enthusiastic about the content of the article. It has the feel of a somewhat superficial literature review that offers little in terms of an new slant or opinion on the current state of the field.

To me, it reads a little like a review article written for the sake of it. It lacks focus, fails to really underscore controversy in the literature, or offer clear opinion on where the field needs to go.

My specific comments are listed below:

In the abstract you state that 'In this review....'. Im not really sure you actually discussed these fundamental processes in great depth. I would suggest revising this statement of your objective here - and also repeat it toward the end of the introduction section. The statement have at the end of the intro section currently reads like the lead-in to an aim and hypothesis for an original article, where a clear statement of the purpose of your review article would be more preferable.

We agree with the reviewer that we lacked some depth in some of our sections, especially when describing these fundamental processes in white adipose tissue. We have added more structure and description of these fundamental processes throughout the review and hope the reviewer finds these changes to be satisfactory.

Related to point one - given that you devote a good amount of space to the impact of bariatric surgery WAT, perhaps your title could be changed to reflect this? i.e., drop the 'with exercise' in the title.

We agree that the impact of bariatric surgery in white adipose tissue was an important issue in the paper, therefore, it was included in the title and the new title is “Targeting white adipose tissue with exercise or bariatric surgery as therapeutic strategies in obesity”.

Many of the sections describe some studies, highlight some inconsistencies in the literature, and then conclude that more work is needed. I urge the authors to offer their opinion of why they think there may be a discord in the literature is some areas of WAT metabolism, and to offer their opinion on where the field should go.

I would encourage you to provide a more robust discussion of the impact of exercise on WAT, and in particular, WAT browning. While there is a growing body of literature on this topic derived from rodent studies - studies in humans are fewer and far less compelling. While rodent WAT can undergo robust browning - developing functional UCP1 - to my knowledge there are no reports to date showing that human WAT has the capacity develop functional UCP1. Perhaps this is the issue - rodent and human WAT are fundamentally different in terms of their ability to undergo browning? I urge you to make your case as to why you think rodent and human studies do not align on this matter - what could be done to reconcile this issue in translatability?

So when mito respiration increases in WAT with weight loss or exercise, how is respiration being normalized? Do you think mito volume/density is increasing per adipocyte? Or is a significant shrinking of the adipocyte lipid droplets responsible for this observation. What is a good marker for mito content in WAT?

In general, we have added more concluding remarks and opinions of where we think the current field is in terms of the effects of exercise on white adipose tissue. We have also included much more information on the effects of exercise on browning of white adipose tissue and how that contrasts between rodents and humans.

In particular, there is some evidence of differences in metabolic responses to exercise between animals and humans, which are probably associated with the brown adipose tissue, specifically related to the functionality, gene, and protein expression. In rodents, the brown adipose tissue depots present in well-defined anatomical location and is homogeneously composed of brown adipocytes; whereas human brown adipose tissue is dispersed and composed by a mixture of white, brown, and brite adipocytes (Liu et al, 2017; Jespersen et al., 2013). Additionally, there are some differences in lipolysis regulation, UCP-1 expression, which are discussed in the manuscript in the “Mitochondrial Content and Capacity” section.

Regarding mitochondrial content in the adipose tissue, the expression of thermogenic and oxidative genes (UCP1, PRDM16, PGC1a, CPT1b, perilipin1 and mt DNA copy number) are good markers for mitochondrial content (Pino et al 2016, Pino et al. 2017, Bosma et al. 2017).  A previous study developed by our research group (Pino et al, 2016) investigated the effects of a 3-week training protocol on markers of mitochondrial respiration and content in the adipose tissue in lean and overweight sedentary individuals. It was found that the training protocol effectively increased the aerobic capacity (ATPmax and VO2peak) but had no impact on mtDNA content and expressions of thermogenic and oxidative genes (UCP1, PRDM16, PGC1a, and CPT1b) in white adipose tissue. However, it is important to highlight that some functional improvement occurred since that important clinical changes in body composition (fat mass decrease) and aerobic capacity markers (increased levels of ATPmax and VO2peak) were observed. The ATPmax of the lean/overweight sedentary participants increased to a level similar to the active participants. It can be suggested that in human, the changes in white adipose tissue are more related to the oxidative capacity then a thermogenic capacity phenotype. In other words, the white adipose tissue can become metabolic more efficient and health, without getting browning or beiging characteristics. Some information was added in the manuscript “Mitochondrial Content and Capacity” section.

We also expanded the section on “Adipocyte Size and Number”, and some information about adipocyte differentiation, lipid deposition and it is effects on energy balance was included. In addition, some information about the positive and negative effects of IL-6 release were added in the “Secretory Function, Tissue Cross-talk and Inflammation” section.

Finally, some discussion about the effects of exercise in changing the metabolic efficiency of the adipose tissue were added in the “White adipose tissue and exercise: endurance vs. resistance training” section.

Your description of recent bariatric surgery data is interesting. Given the impressive - and perhaps most critically - the sustained weight loss achieved by sleeve gastrectomy and bypass procedures, I wonder if these findings reflect what happens in WAT following chronic and sustained weight loss? Whereas perhaps many exercise studies really only reflect and more acute response?

Perhaps it would also be interesting to comment on what real-world value of surgery vs exercise might be given that exercise interventions rarely provide long-term sustained weight-loss.

In general, we have added more opinions about the effects of weight loss provided by exercise or bariatric surgery in the “Exercise, bariatric surgery and adipose tissue” section.

Recent studies have shown that bariatric surgery as an effective option for the treatment of  obesity, since that studies have demonstrated metabolic benefits such as increased insulin sensitivity, modulation of the adipokine secretion and decreased local adipose and systemic inflammation (Coen et al. 2016, Dankel et al, 2011). There were also observed a metabolic impact at the epigenetic level (Benton et al. 2015). For example, intense weight loss and caloric restriction associated with bariatric surgery promotes important changes in the DNA methylation of key genes within adipose tissue (CETP, DNMT3B, FOXP2, HDAC4, KCNQ1, HOX clusters). However, the functional relevance of these changes at the level of DNA methylation have yet to be determined. Future studies focused on the epigenetic regulation of adipocyte function following dramatic weight loss induced by bariatric surgery are on the horizon.

In relation to adipose tissue mitochondrial metabolism, that can be modulated by exercise, a recent study (Hansen et al. 2015) evaluated the adipose tissue mitochondrial respiration and lipolysis following massive weight loss promoted by diet and Roux-en-Y gastric bypass in obese patients with or without type 2 diabetes. It was noted that the mass-specific respiratory capacity of the adipose tissue increased with the weight loss in all patients, the mitochondrial respiratory rates increased and were similar in both groups. Also, the ratio between the maximal capacity of the electron transport system and the oxidative phosphorylation system capacity, lipolysis and insulin sensitivity increased 18 months after the surgery in both groups.

When evaluating respiratory capacity of adipose tissue following such a dramatic loss of adipose mass, the way in which these measurements are normalized becomes critical. For example, mitochondrial content should be considered in the event that the weight loss reduced mitochondrial content. We know from previous studies that mitochondrial content actually decreases with increasing body mass index and fat mass.

However, simply eliminating the adipose tissue (i.e. weight loss, bariatric surgery) is not the only solution to improve metabolic health. In fact, the reduction adipose tissue excess can certainly be beneficial, but it is not obtained through exercise practice alone and usually requires some level of caloric restriction. Changes in life style such as chronic exercise practice associated to health and individually balanced food intake are essential to promote a “healthier” adipose tissue.

Reviewer 2 Report

De Carvalho and Sparks have written a concise, timely and important review article addressing the impact of exercise on white adipose tissue metabolism. For the most part the authors have done a good job in highlighting key findings in the field. With this being said there are several aspects of this work that could be strengthened. These are as follows:

1. When first discussing the impact of exercise on adipose tissue mitochondrial content the authors should cite the first study to demonstrate this phenomenon from Galbo’s laboratory (PMID: 1653528)

2. Line 95”…SREBP1c,…which is a hormone...” This is a transcription factor not a hormone. Please correct.

3. On line 97 the author’s posit that “Targeting adipocyte precursor cells, which ultimately impacts adipocyte differentiation, is an attractive treatment strategy for obesity and metabolic disease”. This statement requires some further contextual detail. Are the authors suggesting that reducing adipocyte differentiation would be an approach to reduce obesity? In the face of positive energy balance would this not lead to increases in ectopic lipid deposition and a worsening of insulin resistance and impairments in glucose homeostasis? Conversely, wouldn’t increasing the number of pre-adipocytes and enhancing differentiation provide a larger sink to “safely” store lipid?

4. On line 114 the authors suggest lipids induce the activation of TLR4. It would be important to state that this likely does not occur directly through an interaction between fatty acids and the TLR4 receptor (PMID 29681442).

5. Beginning on line 130 the authors argue that adipose tissue derived IL-6 can have deleterious effects. While this could certainly be true in some models there is mounting evidence that IL-6 has several positive effects on whole body fuel metabolism. For instance, Peppler et al. (PMID 30383412) recently demonstrated that acute IL-6 treatment enhances liver insulin action in both lean and obese mice. The author’s should consider a more balanced discussion of the role of IL-6 in regulating fuel metabolism.

6. In the discussion of exercise induced changes in adipose tissue mitochondrial biogenesis/browning in humans please reference and discuss a recent paper demonstrating increases in these endpoints in human participants (PMID 30618796).

7. Line 227 the authors refer to a paper demonstrating increases in UCP1 with exercise in human subjects. The referenced paper, #46 in the reference list, is a review paper and it is not clear, at least to this reviewer where this data came from.

Author Response

We would like to thank the Editor and reviewers for their attention and review of our manuscript. All of the reviewers' comments were carefully considered and addressed, and we feel that the manuscript has significantly improved as a result. All changes are indicated with red underline.

De Carvalho and Sparks have written a concise, timely and important review article addressing the impact of exercise on white adipose tissue metabolism. For the most part the authors have done a good job in highlighting key findings in the field. With this being said there are several aspects of this work that could be strengthened. These are as follows:

1.When first discussing the impact of exercise on adipose tissue mitochondrial content the authors should cite the first study to demonstrate this phenomenon from Galbo’s laboratory (PMID: 1653528).

Thank you for your suggestion. Some information was added about the first investigations about the effects of exercise on adipose tissue in the “Introduction” section.

2. Line 95”…SREBP1c,…which is a hormone...” This is a transcription factor not a hormone. Please correct.

Thank you for the correction. The word “hormone” was removed.

3. On line 97 the author’s posit that “Targeting adipocyte precursor cells, which ultimately impacts adipocyte differentiation, is an attractive treatment strategy for obesity and metabolic disease”. This statement requires some further contextual detail. Are the authors suggesting that reducing adipocyte differentiation would be an approach to reduce obesity? In the face of positive energy balance would this not lead to increases in ectopic lipid deposition and a worsening of insulin resistance and impairments in glucose homeostasis? Conversely, wouldn’t increasing the number of pre-adipocytes and enhancing differentiation provide a larger sink to “safely” store lipid?

This is an important point.  We agree that adipocyte differentiation is as important approach to reduce metabolic complications associated with obesity through the expandability hypothesis by Danforth. The growth and differentiation of preadipocytes is controlled by communication between individual cells and/or between cells and the extracellular environment (hormones and growth factors), that affect adipocyte differentiation in a positive or negative manner, and even regulates adipose tissue development and lipid deposition (Gregoire et al, 1998).

Furthermore, both adipocyte hyperplasia and hypertrophy can promote negative effects on lipid metabolism, since that the "flooding" of the adipocytes in the adipose tissue causes a reduction of the blood flow with consequent hypoxia and infiltration of macrophages, and increased lipolysis. Higher levels of free fatty acids in the plasma can lead to a lipotoxicity state (Ghaben and Sherer, 2019; Gregoire et al, 1998). Furthermore, the cytokines produced by macrophages such as TNFα, PAI-1, IL-6, retinol-binding protein 4, MCP-1 and acute phase proteins, can inhibit adipogenesis process (Queiroz et al, 2009).

In face of positive energy balance and increased visceral fat mass, there is a spillover of excessive amounts of lipids which would promote ectopic lipid deposition and the release of inflammatory cytokines, resulting in lipotoxicity in adjacent tissues promoting insulin resistance and impairments in glucose homeostasis (Abildgaard et al, 2018).

The balance of adipocyte size expansion of mature adipocytes and adipogenesis directly affects metabolic health, in this sense, the smaller the adipocytes size lower the susceptibility of developing obesity, diabetes and other metabolic diseases, which supports the importance of increasing the number of pre-adipocytes and enhancing differentiation to provide a “safely” lipid store. Therefore, targeting adipocyte precursor cells, which ultimately impacts adipocyte differentiation, is an attractive treatment strategy for obesity and metabolic disease.

The “Adipocyte Size and Number” session was reformulated and some review about adipocyte differentiation, lipid deposition and it is effects on energy balance were included.

4. On line 114 the authors suggest lipids induce the activation of TLR4. It would be important to state that this likely does not occur directly through an interaction between fatty acids and the TLR4 receptor (PMID 29681442.).

Thank you for the paper suggestion. Some information about the activation of TLR4 was added in the manuscript "Secretory Function, Tissue Cross-talk and Inflammation” section.

5. Beginning on line 130 the authors argue that adipose tissue derived IL-6 can have deleterious effects. While this could certainly be true in some models there is mounting evidence that IL-6 has several positive effects on whole body fuel metabolism. For instance, Peppler et al. (PMID 30383412) recently demonstrated that acute IL-6 treatment enhances liver insulin action in both lean and obese mice. The author’s should consider a more balanced discussion of the role of IL-6 in regulating fuel metabolism.

Thank you for the paper suggestion. Some information about the positive and negative effects of IL-6 release were added in the manuscript in "Secretory Function, Tissue Cross-talk and Inflammation” section.

6. In the discussion of exercise induced changes in adipose tissue mitochondrial biogenesis/browning in humans please reference and discuss a recent paper demonstrating increases in these endpoints in human participants (PMID 30618796).

We would like to report that the suggested paper (PMID 30618796) was previously present in the manuscript but the cited reference was incorrect. The mistake was corrected in the manuscript. Please check page 8, line 351.

7. Line 227 the authors refer to a paper demonstrating increases in UCP1 with exercise in human subjects. The referenced paper, #46 in the reference list, is a review paper and it is not clear, at least to this reviewer where this data came from.

We would like to report that the sentence referred to reference #46 was incorrect. The correct reference is the Otero-Díaz et al (2018) paper, which investigated the effects of 12 weeks of cycling (3 times per week, intensity 70–80% HRmax) in sedentary men and women on markers of metabolism, including UCP1, in the white adipose tissue. The mistake was corrected in the manuscript. Please check page page 8, line 349.

References

Abildgaard, J.; Danielsen, E.R.; Dorph, E.; Thomsen, C.; Juul,A.; Ewertsen, C.; Pedersen, B.K.; Pedersen, A.T.; Ploug, T.; Lindegaard, B. Ectopic Lipid Deposition Is Associated With Insulin Resistance in Postmenopausal Women. J. Clin. Endocrinol. Metab. 2018, 103, 9, 3394–3404.

Benton, M.C.; Johnstone A.; Eccles, D.; Harmon, B.; Hayes, M.T.; Lea, R.A.; Griffiths, L.; Hoffman, E.P.; Stubbs, R.S.; Macartney-Coxson, D. An analysis of DNA methylation in human adipose tissue reveals differential modification of obesity genes before and after gastric bypass and weight loss. Genome Biol. 2015, 16, 8.

Bosma, M.; Minnaard, R.; Sparks, .M.; Schaart, G.; Losen, M.; Baets, M.H.; Duimel, H.; Kersten, S.; Bickel, P.E.; Schrauwen, P.; Hesselink, M.C.K. The lipid droplet coat protein perilipin 5 also localizes to muscle mitochondria. Histochem. Cell Biol. 2012, 137, 205–216.

Coen, P.M.; Goodpaster, B.H. A role for exercise after bariatric surgery? Diabetes, Obes. Metab. 2016, 18, 16–23.

Dankel, S.N.; Staalesen, V.; Bjørndal, B.; Berge, R.K.; Mellgren, G.; Burri, L. Tissue-Specific Effects of Bariatric Surgery Including Mitochondrial Function. J Obesity 2011, Article ID 435245. DOI: 10.1155/2011/435245.

Ghaben, A.L.; Scherer, P.E. Adipogenesis and metabolic health. Nat. Rev. Mol. Cell Biol. 2019. doi: 10.1038/s41580-018-0093-z.

Gregoire, F.M.; Smas, C.M.; Sul, H.S. Understanding Adipocyte Differentiation. Physiol. Rev. 1998, 78, 3.

Hansen, M.; Lund, M.T.; Gregers, E.; Kraunsøe, R.; Hall, G.V.; Helge, J.W.; Dela, F. Adipose Tissue Mitochondrial Respiration and Lipolysis Before and After a Weight Loss by Diet and RYGB. Obesity 2015, 23, 2022–2029.

Jespersen, N.Z.; Larsen, T.J.; Peijs, L; Daugaard, S.; Homøe, P.; Loft, A.; de Jong, J.; Mathur, N.; Cannon, B.; Nedergaard, J.; Pedersen, B.K.; Møller, K.; Scheele, C.  A classical brown adipose tissue mRNA signature partly overlaps with brite in the supraclavicular region of adult humans. Cell Metab. 2013, 17, 798–805.

Liu, X.; Cervantes, C.; Liu, F. Common and distinct regulation of human and mouse brown and beige adipose tissues: a promising therapeutic target for obesity. Protein Cell 2017, 8, 6, 446–454.

Otero-Díaz, B.; Rodríguez-Flores, M.; Sánchez-Muñoz, V.; Monraz-Preciado, F.; Ordoñez-Ortega, S.; Becerril-Elias, V.; Baay-Guzmán, G.; Obando-Monge, R.; García-García, E.; Palacios-González, B.; Villarreal-Molina, M.T.; Sierra-Salazar, M.; Antuna-Puente, B. Exercise Induces White Adipose Tissue Browning Across the Weight Spectrum in Humans. Front. Physiol. 2018, 9, 1781.

Pino, M.F.; Divoux, A.; Simmonds, A.V.; Smith, S.R.; Sparks, L.M. Investigating the effects of Orexin-A on thermogenesis in human deep neck brown adipose tissue. Int. J. Obes. 2017, 41(11):1646-1653.

Pino, M.F.; Parsons, S.A.; Smith, S.R.; Sparks, L.M. Active Individuals have High Mitochondrial Content and Oxidative Markers in Their Abdominal  subcutaneous Adipose Tissue. Obesity 2016, 24, 12, 2467-2470.

Queiroz J.C.F.; Alonso-Vale, M.I.C.; Curi, R.; Lima F.B. Controle da adipogênese por ácidos graxos. Arq. Bras. Endocrinol. Metab. 2009, 53, 5.

Round  2

Reviewer 1 Report

The authors have adequately responded to my original comments.